# Echocardiographic Clustering by Machine Learning in Children with Early Surgically Corrected Congenital Heart Disease

**Wei-Hsuan Chien** [1]  **Cristian Rodriguez Rivero** [2]  **Stijn Daniël Haas** [3]  **Mitchel Molenaar** [3]

## Abstract

The research investigates the time-series clustering from echocardiographic data in children with surgically corrected congenital heart disease (CHD). In recent years, machine learning has been demonstrated to discover sophisticated latent patterns in medical data, yet relevant explainable applications in pediatric cardiology remain lacking. To address this issue, we propose an autoencoder-based architecture to model time-series data with interpretable results effectively. The proposed method outperforms the baseline models in terms of internal clustering metrics. The three clusters also show distinguished differences in patients' outcomes. Patients in Cluster 0 exhibit the poorest prognosis, with an approximate reoperation rate of 40% within the initial six months following the index surgery. The data mining result can potentially facilitate clinicians to stratify patients' prognoses based on echocardiographic and clinical observations in the future.

## 1. Introduction

### 1.1. Complex Congenital Heart Disease

Congenital heart disease (CHD) is fairly common, with a reported prevalence of 7.2 to 8.2 per 1,000 live births in Europe (Van Der Linde et al., 2011). Approximately a quarter of these patients' cardiac structure and function are vastly impaired and life-threatening, also known as critical CHD (Oster et al., 2013). Surgery may be required within the

---

*Equal contribution [1]Institute of Informatics, University of Amsterdam, Amsterdam, Netherlands [2]Department of Computer Science, Cardiff Metropolitan University, Cardiff, United Kingdoms [3]Amsterdam University Medical Center, Amsterdam, Netherlands. Correspondence to: Wei-Hsuan Chien <locriginal@gmail.com>, Cristian Rodriguez Rivero <crodriguezrivero@cardiffmet.ac.uk>, Stijn Daniël Haas <s.d.haas@amsterdamumc.nl>, Mitchel Molenaar <m.a.molenaar1@amsterdamumc.nl>.

*Workshop on Interpretable ML in Healthcare at International Conference on Machine Learning (ICML)*, Honolulu, Hawaii, USA. 2023. Copyright 2023 by the author(s).

first year after birth or even the first days of life. Though advances in healthcare and cardiac surgery have led to an increase in overall survival (Khairy et al., 2010), around 8% of patients were reported to die during the first 18 years of life (McCracken et al., 2018). Additionally, patients growing up after cardiac surgery may have impaired neurocognitive development (Wernovsky, 2006; Martinez-Biarge et al., 2013), reduced exercise capacity (Heiberg et al., 2015), and ultimately decreased quality of life (Ladak et al., 2019; Ringle & Wernovsky, 2016). Hence the need for reoperations cannot be neglected (Monro et al., 2003; Jacobs et al., 2014). The extent of these problems varies greatly among patients, and it is currently difficult for physicians to identify patients at risk. While currently lacking a robust tool, clustering or classification models may assist physicians in identifying children at risk of these long-term outcomes and provide them with tailored care to individual patients' needs. Conventionally, echocardiographic parameters have been used as a predictor and shown to have certain prognostic values in children with CHD (Patel et al., 2010; Prota et al., 2019). However, due to the heterogeneity of CHD pathogenesis, no echocardiographic parameter is a predictor to monitor prognosis across all diseases. Furthermore, conventional approaches rely on the contemporary classification of CHD, which is based on the cardiac structure at birth and may not adequately capture heterogeneous progression patterns.

### 1.2. Machine Learning in Medical Research

Machine learning (ML) has gained significant attention in medical research due to its potential to improve diagnostics, risk stratification, and prognosis prediction in cardiovascular disease (Gandhi et al., 2018; Seetharam et al., 2020). Supervised ML enables the prediction of specific labels, while unsupervised ML allows data exploration without labels. Clustering, an unsupervised learning technique, has been applied to identify patient subgroups with similar clinical features and disease prognosis (Basile & Ritchie, 2018; Quer et al., 2021). This knowledge can assist physicians in making more informed clinical decisions and optimising therapeutic strategies. Moreover, it provides a framework for precision medicine, facilitating the exploration of sophisticated patterns and the identification of patient similarities (Quer et al., 2021). In the scope of cardiovascular diseases,

cluster analysis often involves clinical and cardiovascular imaging variables. For example, in the work of Monika et al. (Przewlocka-Kosmala et al., 2019), the team applied clustering in patients with heart failure with preserved ejection fraction (HFpEF), and the result separated three subgroups characterised by a relatively isolated impairment of left ventricular systolic reserve. Another research differentiated four clusters in patients with left ventricular diastolic dysfunction (LVDD), and their clustering model performed better at predicting event-free survival than guideline-recommended algorithms (Lancaster et al., 2019). They also suggested the natural patterns of clustering may help eliminate indeterminate results and improve clinical outcome prediction. Horiuchi et al. used clinical data from patients with acute heart failure (ACF) and differentiated three clusters with a K-means clustering (Horiuchi et al., 2018). The research suggested the heterogeneity in this population and that a uniform treatment therapy might not fit all individuals. In a recent work of Hoeper et al. (Hoeper et al., 2020), a hierarchical agglomerative clustering algorithm was performed to classify idiopathic pulmonary arterial hypertension (IPAH). They noted that the various clusters resulted in different survival rates. Although the research also suspected these could be attributed to differences in etiology and pathophysiology, the need for advanced clinical testing (e.g. imaging) to define phenotypes for clinical decision-making was also accentuated (Badagliacca et al., 2020). To reduce feature dimensions, some researchers have investigated vector embeddings using data from heart failure patients (Choi et al., 2016; Denaxas et al., 2018). Oliver et al. utilised an extension of deep embedding clustering approaches to stratify the risk in heart failure cohorts in both an unsupervised and semi-supervised fashion (Carr et al., 2020).

However, due to the unsupervised nature and reliance solely on unlabeled data, the major limitation to applying clustering in medical data is the lack of interpretability for clinicians. In addition, conventional clustering algorithms often treat input variables as static data, which may not adequately capture the dynamics of disease progression over time, as time-dependent variables in medical data are often correlated non-linearly. In the context of time-series clustering, the high dimensionality and noise in time-series data pose challenges for conventional clustering algorithms.

In light of those mentioned above, we propose a robust architecture to identify the risk profile of children with surgically corrected CHD. This involves constructing an ML model to cluster these patients into distinct phenotype groups with different clinical outcomes. The integration enhances the interpretability of the clustering results. By incorporating this additional information, we strive to bridge the gap between machine learning techniques and clinical understanding, enhancing the practical utility of our findings and allowing for early risk identification and personalised treatment for patients.

## 2. Related Work

Much of the existing research in data science has focused on an effective approach to represent time-series data. The two major categories in time-series clustering are raw-data-based models and feature-based models.

Raw-data-based models perform clustering algorithms directly on raw data, where the distance measured between time-series is mainly modified with scaling or distortion. These methods match two time-series according to their shapes, measured by a non-linear stretching and contracting of the time axes. For example, dynamic time warping (DTW) minimises the pairwise Euclidean distance by mapping two points in time-series (Berndt & Clifford, 1994). Petitjean et al. proposed DTW Barycenter Averaging (DBA) to combine K-means and dynamic time-wrapping for better alignment (Petitjean et al., 2011). K-Spectral Centroid (K-SC) is a similarity metric invariant to scaling and shifting to explore the dynamics in time series (Conan-guez et al., 2018). K-shape is a modification of K-means, which measures the distance based on the cross-correlation of two time-series (Paparrizos & Gravano, 2015). The partitioning approach works well with low-dimensional, well-separated data, whereas time-series data are often multidimensional with intersections and embedded clusters. Also, since all data points are measured to assign clusters, they are sensitive to outliers, noise, and extraneous data, impacting the clustering accuracy.

Unlike raw-data-based models, feature-based models focus on extracting feature representations from input data to a lower dimension. These methods vary in effectively performing dimension reduction and selecting an appropriate similarity measure. Because data points belonging to the same cluster can be rather distant in a high-dimensional space, we often lose meaningful differentiation between similar and dissimilar objects when the number of dimensions increases (Assent, 2012). Since multivariate time-series data often inherits high dimensionality, feature-based models aim to generate an effective latent representation before or when performing a clustering algorithm. These techniques transform raw input data into a set of vectors in a latent space, also known as vector embedding or latent representation. This process can reduce the original inputs' dimensionality while preserving important features.

Principle Component Analysis (PCA) is commonly used to find a lower-dimensional embedding. Due to the nature of PCA, it is more appropriate to subtract features with linear relationships (Camacho et al., 2010), while time-series tend to be non-linear and have more complicated representations. On the other hand, autoencoder has been popularly applied

in time-series data along with the development of deep learning (Vincent et al., 2010). An autoencoder is a feedforward, non-recurrent neural network comprising an encoder and a decoder. Since an autoencoder is trained to optimise a reconstruction loss by minimising the error between the input and the reconstructed output, the products forwarded by the encoder can therefore be regarded as a lower-dimensioned latent representation of the input data.

To illustrate, Deep Temporal Clustering (DTC) was first introduced by Sai et al. (Madiraju et al., 2018), which leverages a temporal autoencoder to learn non-linear features and a clustering layer to assign clusters by measuring the KL divergence. Ma et al. (Ma et al., 2019) proposed the Deep Temporal Clustering Representation (DTCR), which integrates temporal reconstruction and the K-means objective into a seq2seq model. This approach leads to improved cluster structures and thus obtains cluster-specific temporal representations. Dino et al. proposed Deep Time Series Embedding Clustering (DeTSEC), which firstly utilises a recurrent autoencoder to produce a preliminary embedding representation and assign clusters with a clustering refinement stage (B, 2020). Lee et al. (Lee & Schaar, 2020) proposed a deep learning architecture for clustering time-series data, where each cluster comprises patients who share similar future outcomes, including adverse events and the onset of comorbidities. They later introduced another architecture that integrates outcome-oriented phenotyping of clinical pathways (Lee et al., 2020).

# 3. Methodology

## 3.1. Datasets

We conducted a retrospective single-centre study at the Emma Children's Hospital of the Amsterdam University Medical Center (AUMC) on pediatric patients with CHD who underwent early surgical correction. The study utilised echocardiographic data stored on EchoPac v203, following the American Society of Echocardiography guidelines (Lai et al., 2006). The study included patients whose first surgery on cardiopulmonary bypass was set as the index operation. Echocardiographic parameters with less than 20% missing values were analysed. In contrast, patients with less than five consecutive echocardiographic exams or a total follow-up shorter than 12 months were excluded from the study. The anonymised data from 182 patients included clinical variables, echocardiographic measurements, and outcomes, such as time-to-reoperation, observed over an average of 80 months.

External validation was done using a second openly available dataset from Physionet (Reyna et al., 2020). This dataset comprises time-dependent variables, measured hourly, and outcomes defined as time-to-event (occurrence of sepsis). The dataset includes 20,336 intensive care unit (ICU) patients, of which we selected a subset of 363 patients with enough time-series data on 13 different variables to make the data structure compatible with our experimental setting.

## 3.2. Data Preparation

We applied an uneven average aggregation method to handle the varying intervals between consecutive echocardiograms during follow-up for patients with CHD. This method effectively decreases the number of missing values by aggregating time-dependent variables based on their measurement frequency. In the first six months after the index operation, where patients were monitored more frequently, the observation values were averaged on a monthly interval. From the seventh month onwards, intervals of three, six, and 12 months were used for aggregation. Simple linear imputation and backward-filling methods were employed for imputing the remaining missing data, as they are more clinically interpretable from clinicians' perspectives.

We focused on the first 72 months after cardiac surgery and selected 14 time-dependent features for clustering based on expert consultation. These features include body surface area (BSA), weight, left fractional ventricular shortening (%FS), aortic valve peak velocity (AV Vmax), aortic valve peak gradient (AV maxPG), interventricular septum diastolic thickness (IVSd), interventricular septum systolic thickness (IVSs), left ventricular internal diastolic dimension (LVIDd), left ventricular internal systolic dimension (LVIDs), left ventricle posterior wall diastolic thickness (LVPWd), left ventricle posterior wall systolic thickness (LVPWs), pulmonary valve peak velocity (PV Vmax), and pulmonary valve peak gradient (PV maxPG).

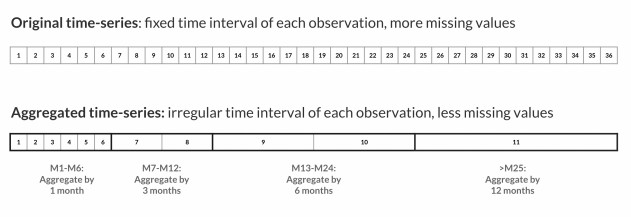

*Figure 1.* Time steps aggregation

## 3.3. Model Implementation

### 3.3.1. BASELINE MODELS

To better compare the performance of our proposed method, we chose K-means, DBA, and K-shape as our baseline models. These methods are raw-data-based clustering techniques and require a prespecified hyperparameter k value. K-means

is a naive baseline since it directly measures the Euclidean distances among observations. Both DBA and K-shape apply certain distortions to align two time-series better, and they are therefore deemed more robust than K-means in time-series clustering (Petitjean et al., 2011; Paparrizos & Gravano, 2015).

### 3.3.2. PROPOSED METHOD

Motivated by the success in applying deep learning, we propose an autoencoder-based deep neural network architecture to better model this problem. The autoencoder acts as the backbone of our proposed method. Since the latent representation does not necessarily form clusters with well-separated outcomes, we added a predictor to facilitate feature learning. This allows the encoder to play the most important role in generating a well-represented vector space; adding the predictor can force the encoder to form vectors embedded with outcome signals. This adapts the embedding to the problem of interest.

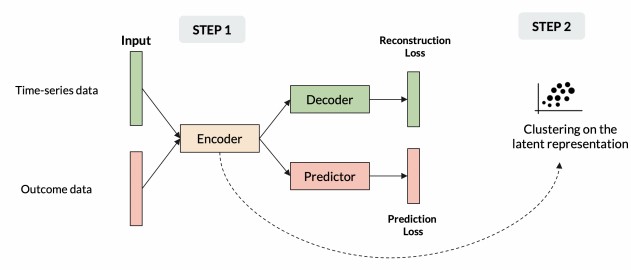

*Figure 2.* Overall process of the proposed method

The proposed method reduced input dimensions to a feature space before applying a clustering algorithm. First, we trained the autoencoder-based model with two tasks: the decoder focuses on reconstructing the latent representation into the original time series, and the predictor focuses on predicting patients' outcomes. The total loss is therefore contributed by two tasks, a reconstruction loss and a prediction loss. After training the model, the whole dataset was fed to the encoder to generate the latent representation from the input. Clusters for each patient were subsequently assigned by using a standard clustering algorithm K-means on the embedded space. Figure 2 illustrates how clustering is formed using the proposed architecture.

Focusing on the model architecture in more detail: the encoder and decoder are formed of Long Short-Term Memory (LSTM), a recurrent neural network (RNN) deemed robust for sequential data. Unlike standard feedforward neural networks, LSTM has feedback connections, and it can not only process single data points but also entire sequences of data. LSTM is known for its memory capacity, meaning that it can memorize previously seen data. Three different

gates realize the characteristic of storing information over a period of time: a forget gate, an input gate, and an output gate. Each gate decides whether and how strongly the previous signal can be passed to the current and next states. The predictor is made up of several layers of the Convolutional Neural Network (CNN). CNN models learn an internal representation of inputs from different dimensions in a process referred to as feature learning. The same process can be harnessed on one-dimensional time series data, leveraging the filtered features to predict outcomes. The overall details of respective layers can be found in Appendix, Figure 4.

### 3.4. Experimental Setup

Before clustering, all the input variables were normalized to improve the efficiency of clustering algorithms and accelerate the training. The final input was reshaped into a three-dimensional matrix. We used an elbow method and an average Silhouette width to determine the optimal k value for clustering, which have been commonly utilized as a criterion for selecting the number of clusters. The k value was found to be three and used in our analysis.

The autoencoder-based architecture is optimized through back-propagation. The input first passes forward through the network to yield a reconstructed time series and the predicted outcome. After comparing the predictions with the original time series and ground truth, the architecture updates its parameters based on the computed loss over the whole network. The total loss is composed of the reconstruction loss and the prediction loss, multiplied by their respective weights: $Loss_{total} = loss_{reconstruction} * w_1 + loss_{prediction} * w_2$

The AUMC and external datasets were randomly split into training, validation, and test sets. The total training was set to be 200 epochs, and the training terminated early if the validation loss did not improve for 20 epochs to prevent overfitting. At each epoch, the training data was fed to the model, and the model updated the trainable parameters to minimize the loss.

### 3.5. Evaluation Metrics

The clustering results are generally assessed by clustering validation indexes (CVI). The external indexes require the externally supplied class labels to measure the similarity of formed clusters, while internal indexes evaluate the clustering structure itself without external labels or information. Given that the external ground-truth labels are not attainable in our research, the clustering results are evaluated by the internal indexes: Silhouette score and COP index.

### 3.5.1. SILHOUETTE SCORE

The Silhouette score is a widely-used internal metric that measures the similarity between each sample and its assigned cluster (the intra-cluster distance) compared to other clusters (the nearest-cluster distance) (Rousseeuw, 1987). The silhouette ranges from $-1$ to $+1$, where a high value indicates that the object is well matched to its own cluster and poorly matched to neighbouring clusters. Values near 0 indicate overlapping clusters. Negative values generally indicate that a sample has been assigned to the wrong cluster, as a different cluster is more similar.

For data point $i$ in the cluster $C_i$ ($i \in C_i$), the mean distance between $i$ and all other data points in the same cluster is denoted as $a(i)$, where $d(i, j)$ is the distance between data points $i$ and $j$ in the cluster.

$$a(i) = \frac{1}{|C_i| - 1} \sum_{j \in C_i, i \neq j} d(i, j)$$

The smallest mean distance of data point $i$ to all points in any other cluster, which the sample is not a part of, is denoted as $b(i)$. The cluster with the smallest mean dissimilarity could be interpreted as the nearest cluster since it is the next best-fit cluster for sample $i$.

$$b(i) = \min_{k \neq i} \frac{1}{|C_k|} \sum_{j \in C_k} d(i, j)$$

The silhouette value for each data point $i$ is therefore defined as follows:

$$s(i) = \frac{b(i) - a(i)}{\max\{a(i), b(i)\}}, if\ |C_i| > 1$$

### 3.5.2. COP INDEX

The COP-index is the ratio of the tightness within the cluster to the farthest adjacent distance (Gurrutxaga et al., 2010). It can assess the quality of clustering results with an estimate of the intra-cluster variance (cohesion) in the numerator and an estimate of the inter-cluster variance (separation) in the denominator.

Given a dataset, $X = \{x_1, x_2, ... x_N\}$, a cluster $C$ is a subset of data points in the dataset. The centroid of a cluster is denoted as $\overline{C}$, whereas $P^Y = \{C_1, C_2, .. C_k\}$ is a set of disjoint clusters of a subset of the dataset.

$$COP(P^Y, X) = \frac{1}{|Y|} \sum_{C \in P^Y} |C| \frac{intra_{COP}(C)}{inter_{COP}(C)}$$

where

$$intra_{COP}(C) = \frac{1}{|C|} \sum_{x \in C} d(x, \overline{C})$$

$$inter_{COP}(C) = \min_{x_i \notin C} \max_{x_j \in C} d(x_i, x_j)$$

### 3.6. Statistical Analysis

The association across clusters is examined by descriptive statistics. The mean fluctuations of continuous variables, namely the difference between the maximum and minimum observed values, are used to determine whether there are any statistically significant differences within clusters with a one-way ANOVA test. Categorical variables are described as percentages and compared by the Pearson chi-square test. We investigate the outcomes of patients with Kaplan-Meier curves and the log-rank test to assess the association between clusters and corresponding prognosis.

## 4. Result

### 4.1. Model Evaluation

To inspect the performance of the autoencoder, we used a heatmap to visualize the original input and reconstructed output. Generally, similarity can be observed between the input time series and reconstructed ones. Along with more training epochs, the reconstructed time series more closely resembled the inputs (Appendix, Figure 5). This demonstrated that the encoder was able to deconstruct the original inputs into a well-represented vector space; meanwhile, the decoder could effectively reconstruct the vectors back to the original inputs.

In terms of internal validation indexes, we repeated all the clustering methods iteratively thirty times with thirty different random seeds to decrease output variances. All the clustering results were evaluated with the Silhouette value and COP index at each iteration. The resulting scores were averaged and displayed in Table 1, accompanied by their standard error. We can therefore see that the proposed method outperforms all the baseline models, namely K-means, DBA, and K-shape, indicating that our proposed method is able to form clusters with greater quality than others. These results were consistently validated in both the AUMC and external datasets. The highest average Silhouette score was found in the proposed method regardless of the datasets (AUMC dataset: 0.46, External dataset: 0.75). The minimal COP index was also seen in the proposed method, 0.11 for the AUMC dataset and 0.12 for the external one. Finally, we chose the model with the greatest Silhouette score to assign cluster labels and performed cluster analysis.

### 4.2. Cluster Analysis

Our proposed method divided the patients into three clusters. Cluster 1 accounted for the majority of the research cohort (n=100), while 48 patients were assigned to Cluster 2, and the remaining 34 patients belonged to Cluster 0. The

| Models | AUMC Dataset | | | | External Dataset | | | |
| --- | --- | --- | --- | --- | --- | --- | --- | --- |
| | Silhouette Score | | COP Index | | Silhouette Score | | COP Index | |
| | avg | std | avg | std | avg | std | avg | std |
| K-means | 0.37 | 0.08 | 0.21 | 0.01 | 0.36 | 0.03 | 0.31 | 0.01 |
| DBA | 0.41 | 0.08 | 0.22 | 0.01 | 0.36 | 0.02 | 0.31 | 0.01 |
| K-shape | 0.43 | 0.07 | 0.22 | 0.01 | 0.27 | 0.07 | 0.42 | 0.03 |
| Proposed method | 0.46 | 0.15 | 0.11 | 0.01 | 0.75 | 0.07 | 0.12 | 0.03 |

*Table 1.* Performance comparison on the AUMC dataset and the external dataset

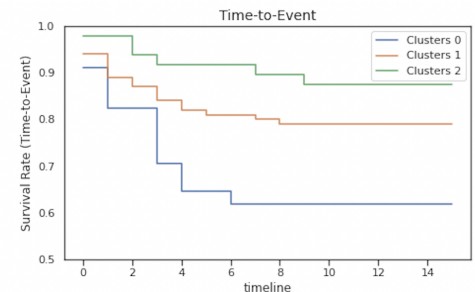

*Figure 3.* Survival free of reoperation across clusters

line plot in Appendix, Figure 6 provides us with a visual impression of the temporal trajectories across clusters. The overall trend in respective clusters seemed to be similar, yet the trajectories in Cluster 0 were more fluctuant along with the time progression, especially in AV Vmax and AV max PG. Cluster 1 and 2 changes were relatively small, resulting in smoother temporal curves. It is also noteworthy that the pattern could be sensitive to some extreme individuals considering the limited size of patients in each cluster. With that being said, in a setting with a larger research population, the temporal clustering could yield more compelling results.

In the results, certain variables showed significant fluctuations: BSA, AV Vmax, AV maxPG, and IVSd. The heatmap in Appendix, Figure 7 visualizes the relative change in three clusters. We can observe that Cluster 0 had the greatest fluctuations, shown in a lighter colour, while the temporal change in Cluster 2 seemed to be less noticeable than in Cluster 1, depicted in a darker colour. Regarding categorical variables, in addition to the features chosen for clustering, six frequently seen preliminary diagnoses were also taken into the post-hoc analysis, including anomalous pulmonary venous drainage (APVD), atrioventricular septal defect (AVSD), transposition of the great arteries (TGA), tetralogy of Fallot (TOF), ventricular septal defect (VSD), and other diagnoses (Appendix, Table 2). In general, we did not observe any remarkable cluster differences in gender and most of the disease diagnoses. VSD, however, was the only diagnosis that is significantly less observed in Cluster 0.

Lastly, we observed an outcome distinction in three clusters

in the Kaplan–Meier curve. The outcome was predefined as event-free survival, and an event refers to receiving a reoperation. According to Figure 3, patients in Cluster 0 seemed to have the worse outcome. Around 40% of patients experienced reoperations within the first six months after the index surgery, whereas the figures for Cluster 1 and Cluster 2 were less than 20% and 10%, respectively. Using a multivariate log-rank test, the result also showed that at least one group's outcome differed from the others (statistics: 8.11, p-value: 0.02).

## 5. Discussion and Future Work

Routine echocardiography is the most common test used in pediatric cardiology. This regular check-up can facilitate clinicians in evaluating treatment efficacy or disease progression. However, the heterogeneity of congenital heart disease may complicate this task. Unsupervised machine learning algorithms could therefore be helpful in data mining. Since cardiac structures and functions tend to develop non-linearly, the echocardiographic observations might also inherit this non-linearity (Lopez et al., 2017). Autoencoder, in particular, is useful to extract features when dealing with non-linear data. In addition, autoencoders are effective even with a small dataset (Zhou et al., 2017). Although researchers have leveraged different autoencoder-based models for time-series clustering, applied value in medical research remains unexplored, especially in pediatric patients with congenital heart diseases. To mitigate the research gap, we propose an autoencoder-based architecture that could be regarded as the optimal approach in our research. The autoencoder-based model can effectively address high dimensionality and non-linearity in temporal data. Also, adding the predictor enhances feature learning by jointly optimizing the loss. This architecture is demonstrated to extract features effectively and validate them to form well-separated clusters in the internal and external datasets.

Although the limited size of the datasets might not allow us to draw any firm conclusion, some insights in our data mining could potentially offer a promising direction. The clinical interpretability of the clustering result and their respective constraints are discussed in the following sections.

We also introduce some state-of-the-art frameworks, which may be worth further investigation in the future.

### 5.1. Latent Causal Relationship

According to the survival analysis and post-hoc analysis, the clustering result seems to imply that patients who experienced greater cardiac function and structure changes are associated with a higher incidence of reoperation. Unlike chronic heart failure, in which left ventricular ejection fraction (LVEF) plays an essential role in assessing prognosis, there is not a prominent echocardiographic parameter serving as a key predictor across all congenital heart diseases. Our research shows that the differences in the aortic valve peak gradient (AV Vmax) and the aortic valve peak velocity (AV maxPG) are the most significant parameters among clusters. Since our method extracts the essential features from the original data, we could deduce that these two parameters might contribute the most in defining clusters.

However, we cannot assure the latent causal relationship between the temporal changes in echocardiographic parameters and reoperation. Although it is known that the reoperations will influence the parameters in the current setting, we are not sure how exactly the reoperation might change individuals' cardiac structures and functions. With that being said, we cannot exclude the possibility that the reoperation might alter cardiac functions and structures directly and indirectly contribute to the changes in echocardiographic parameters. Since the clustering considers both the measurement before and after reoperations, the clustering results could be attributed to the reoperation. Namely, it is possible that the model assigns patients to the same clusters because they receive similar surgical reoperations. A possible approach to model this issue is to set the time of the event as a research endpoint and discard the observations onwards. We should be careful about this approach since dropping off all the observations after the event might lead to variable-length data, as the proposed architecture requires fixed-length time-series inputs. When considering most of the reoperation occurs in the early stages, we might encounter a great loss in data, which can lead to instability and inconsistency in clustering. Another possible approach is redefining the outcome of the event after the observed period. In our case, the new outcome could be defined as either reintervention or reoperation after 72 months and transform any event within the observed period as a binary variable. In this way, any reoperation or reintervention can serve as dependent variables which contribute to both the phenotyping and future prognosis prediction. The other way is to divide the overall observation duration into two periods. In the first period, we could use the observations within the first two years after the index surgery to assign clusters. After that, we can then utilize the outcome data in the succeeding years to analyze the differences in event-free survival.

### 5.2. Missing Data Imputation

The presence of missing values frequently arises in medical research, which throws hidden challenges to data mining and analysis. The aggregated time steps we used are a compromised trade-off for the decrease in missing values; however, this modelling might lead to a loss in temporal information and some fine dynamics of parameter changes. Especially for the measurements after the second year, since the aggregated interval increases to 12 months, some informative details might be neglected. We chose a simple linear imputation when imputing the remaining missing values, even though the development of dimensions and structure is likely non-linear. Hence, the limitations of the missing data imputations should be considered in interpreting the results. The amount of imputed data could lead to inappropriate distance measurement and impact clustering.

To further improve the quality of imputed missing data as well as the clusters' results, a Generative Adversarial Network (GAN) can be considered. GAN is a deep learning framework that learns the latent distribution of a dataset and can subsequently synthesize samples from random (Goodfellow et al., 2014). In the field of multivariate time-series imputation, Luo et al. (Luo et al., 2018) proposed a GAN for data augmentation, which employs a Gated Recurrent Unit (GRU) to address the temporal irregularity of the incomplete time series. ClusterGAN is brought out by Mukherjee et al. (Mukherjee et al., 2019) to achieve clustering in the latent space. Their method samples latent variables and is coupled with an inverse network trained jointly with a clustering-specific loss. This model can preserve latent space interpolation across categories, even though the discriminator is never exposed to such vectors.

## 6. Conclusion

This is the first study that assesses an autoencoder-based method for time-series clustering on echocardiographic data of children with surgically corrected CHD. Machine learning algorithms have been applied widely in medical research, and cluster analysis is used in phenotypes detection to understand the future prognosis and improve tailored care. However, most existing applications focus on clustering static data, which neglects the temporal pattern of echocardiographic and clinical variables. Time-series data's inherent high dimensionality and noise often fail conventional clustering methods. Furthermore, raw-data-based time-series clustering methods do not necessarily divide clusters with distinct outcomes. We propose a robust architecture to mitigate current research gaps to extract features from time-series data to a vector space before applying a partitioning clustering algorithm. The proposed method successfully outperforms the baseline models in both the Silhouette score (0.46 in the AUMC dataset, 0.75 in the external dataset) and

COP index (0.11 in the AUMC dataset, 0.12 in the external dataset) and identifies three distinct outcome-distinctive clusters. Patients in Cluster 0 are associated with the worst prognosis, with approximately 40% undergoing reoperations within the first six months after the index surgery. In contrast, the reoperation rates for Cluster 1 and Cluster 2 were less than 20% and 10%, respectively. The multivariate log-rank test confirmed significant outcome differences among the clusters (statistics: 8.11, p-value: 0.02). The post-hoc analysis also shows that the temporal pattern of AV Vmax, AV maxPG, and IVSd might be the most important echocardiographic parameters in defining patients' clusters. Despite the promising results, several limitations should be considered when applying the result in clinical practice. The latent relationship between echocardiographic trajectories and reoperation could potentially influence the clustering result, and we can consider remodelling the outcome to address this issue. The considerable amount of missing data and simple linear imputations can also influence clustering performance, while several promising frameworks using generative adversarial networks should be considered in future work.

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

## A. Appendices

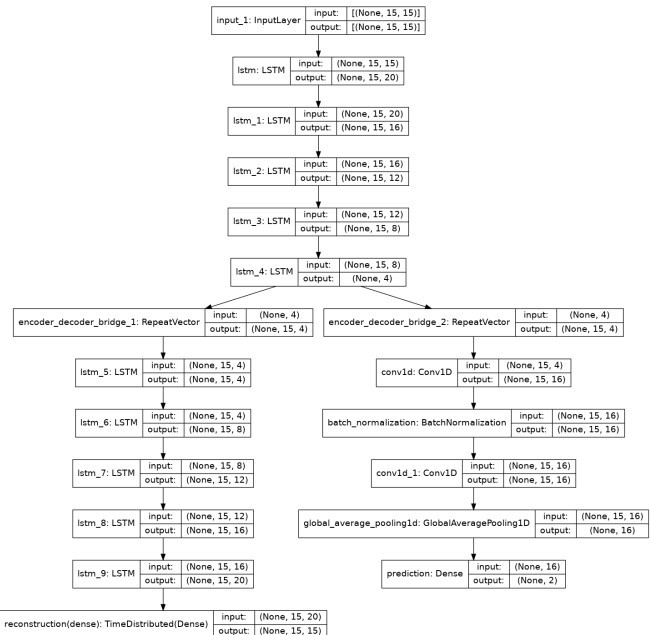

*Figure 4.* Details of the autoencoder-based model

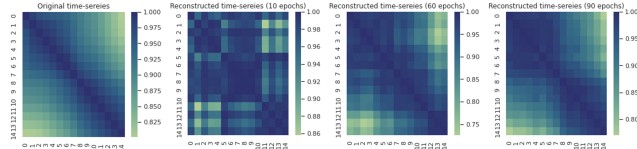

*Figure 5.* Input and reconstructed output

**Echocardiographic Clustering by Machine Learning in Children with Early Surgically Corrected Congenital Heart Disease**

| Variables | Cluster 0 (n=34) | Cluster 1 (n=100) | Cluster 2 (n=48) | statistics | p-value |
|---|---|---|---|---|---|
| BSA | 0.49 (±0.16) | 0.48 (±0.12) | 0.42 (±0.19) | 3.14 | 0.05* |
| Weight | 15.72 (±12.82) | 15.27 (±9.49) | 15.90 (±19.93) | 0.04 | 0.96 |
| %FS | 22.51 (±18.85) | 17.07 (±8.36) | 19.53 (±11.35) | 2.81 | 0.06 |
| AV Vmax | 1 (±1.01) | 0.6 (±0.55) | 0.47 (±0.36) | 7.48 | 0.00*** |
| AV maxPG | 16.56 (±23.99) | 7.37 (±12.52) | 4.80 (±4.98) | 7.34 | 0.00*** |
| IVSd | 3.84 (±3.17) | 2.86 (±1.17) | 2.98 (±1.86) | 3.5 | 0.03* |
| IVSs | 4.57 (±2.32) | 4.18 (±1.68) | 3.87 (±1.43) | 1.59 | 0.21 |
| LVIDd | 16.65 (±7.62) | 15.76 (±4.61) | 14.49 (±5.94) | 1.53 | 0.22 |
| LVIDs | 11.05 (±5.32) | 11.07 (±3.67) | 10.20 (±5.50) | 0.63 | 0.53 |
| LVPWd | 3.55 (±4.17) | 2.56 (±1.06) | 2.85 (±1.55) | 2.74 | 0.07 |
| LVPWs | 4.95 (±2.19) | 4.6 (±1.54) | 4.52 (±1.92) | 0.63 | 0.53 |
| PV Vmax | 1.85 (±1.32) | 1.62 (±1.1) | 1.30 (±1.1) | 2.37 | 0.10 |
| PV maxPG | 41.88 (±35.39) | 35.37 (±31.86) | 25.16 (±30.28) | 2.88 | 0.06 |
| Sex | 0.65 | 0.65 | 0.60 | 0.72 | 0.95 |
| APVD | 0.00 | 0.01 | 0.04 | 2.71 | 0.26 |
| AVSD | 0.10 | 0.09 | 0.09 | 0.02 | 0.99 |
| TGA | 0.55 | 0.19 | 0.23 | 5.93 | 0.05 |
| TOF | 0.21 | 0.45 | 0.26 | 3.21 | 0.20 |
| VSD | 0.00 | 0.25 | 0.33 | 9.48 | 0.01** |
| Other diagnosis | 0.62 | 0.32 | 0.30 | 3.05 | 0.22 |

*Table 2.* Clinical characteristics in respective phenogroup

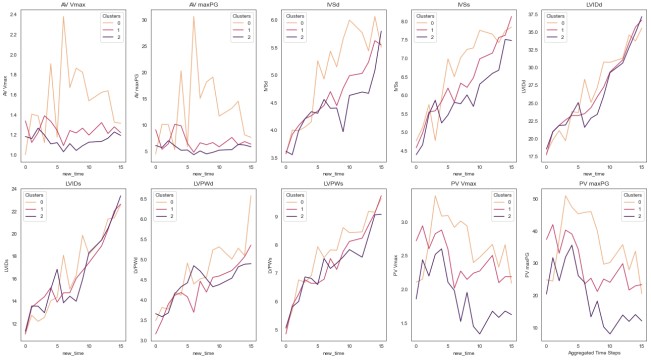

*Figure 6.* Temporal trajectories of echocardiographic indices

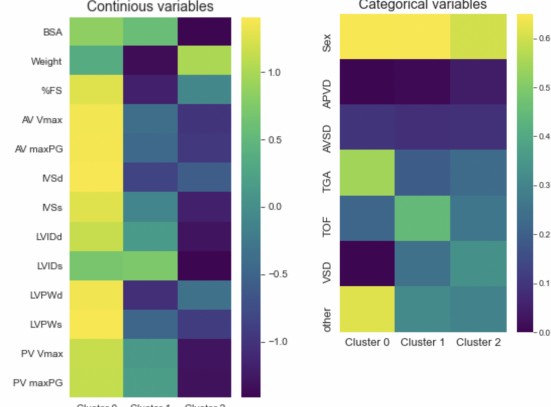

*Figure 7.* Visualization of variables for three clusters

