# OpenReview forum: "Echocardiographic Clustering by Machine Learning in Children with Early Surgically Corrected Congenital Heart Disease"
_ICML.cc/2023/Workshop/IMLH — IMLH 2023 Oral_

### Official Review · Reviewer_mvAR · 2023-06-15
**A well-described work with clinical relevance**

**Rating:** 6
**Confidence:** 3

**Review:**

This manuscript presents an application of auto-encoder-based clustering for the time series echocardiographic data, with the aim to predict the patient prognosis.

Pros:

The application is clinically relevant and the data modality is new (time-series echocardiographic data).
The clinical background and the related works are introduced in detail.
Comprehensive analysis is performed for interpretating the experiment results.

Cons:

Visual examples of raw echocardiographic data are expected to be presented in the manuscript. This may facilitate readers who are not familiar with this modality to better understand the task.
The manuscript can be made more concise such that additional analysis can be moved to the main text from the appendix.

---

### Official Review · Reviewer_EdFp · 2023-06-16
**This paper presents a robust method for survival analysis in time-series data.**

**Rating:** 7
**Confidence:** 3

**Review:**

The paper is well-written and presents an autoencoder-based method for time-series clustering on echocardiographic data. The proposed method outperforms the baseline models in terms of internal clustering metrics. Meanwhile, the paper leaves two issues of latent causal relationship and missing data imputation for future research.

The problem with this paper is the lack of sufficient explanation of the missing data imputation. It would be helpful if the authors could explain the details of the method they chose to impute missing data and provide the missing rate of the data. Moreover, the authors could consider using non-linear methods for missing data imputation to demonstrate the negative impact of linear imputation on their method. This is just a suggestion.

---

### Official Review · Reviewer_dSie · 2023-06-18
**Overall a detailed and well-organized long-research paper**

**Rating:** 7
**Confidence:** 4

**Review:**

Cluster analysis has been widely used in many medical research. Early diagnosis of CHD with the machine learning model is an interesting research topic. Please revise your paper based on the following suggestions:
1. Please ensure to clearly state your research topic and discuss the advantages and disadvantages of cluster analysis in your introduction.
2. Could you elucidate how machine learning methods enhance cluster analysis and elaborate on its benefits in your introduction?
3. Could you clarify what you mean by "we aim to..."?
4. In the section on related work, you delve into two main categories in time series clustering. However, from my perspective, there's no need to be overly verbose here. Instead, highlight the differences between these two models and compare them with your proposed model for a more fitting approach.
5. You've stated that 13 different variables will be used to make the data structure compatible with your experimental setting. Could you detail which variables you used, and explain the process for parameter selection? Is it a manual process?
6. You've noted a distinct outcome in three clusters in the Kaplan-Meier curve. Could you shed some light on its clinical significance?
7. Please include quantitative data in your abstract and conclusion sections to substantiate your findings.

---

### Meta-Review · Area_Chair_o7V7 · 2023-06-19

**Recommendation:** Accept (Oral)
**Confidence:** 4

**Metareview:**

Reviewers are generally positive in recommending the acceptance of this manuscript but also raise concerns. Please address them in the final version.

---

### Decision · Program_Chairs · 2023-06-20

Accept (Oral)